# Within and between-day variation and associations of symptoms in Long Covid: Intensive longitudinal study

Christopher Burton[1]◉*, Helen Dawes[2]◉, Simon Goodwill[3], Michael Thelwell[3], Caroline Dalton[3]◉

1 Academic Unit of Primary Medical Care, University of Sheffield, Sheffield, United Kingdom, 2 College of Medicine and Health, University of Exeter, Exeter, United Kingdom, 3 Advanced Wellbeing Research Centre, Sheffield Hallam University, Sheffield, United Kingdom

◉ These authors contributed equally to this work.
* chris.burton@sheffield.ac.uk

**Data Availability Statement:** "The full dataset for this paper comprising baseline survey, app data and activity summary statistics are on the Figshare online research data archive (ORDA) of the

## Abstract

### Background

People with Long Covid (Post Covid-19 Condition) describe multiple symptoms which vary between and within individuals over relatively short time intervals. We aimed to describe the real-time associations between different symptoms and between symptoms and physical activity at the individual patient level.

### Methods and findings

Intensive longitudinal study of 82 adults with self-reported Long Covid (median duration 12–18 months). Data collection involved a smartphone app with 5 daily entries over 14 days and continuous wearing of a wrist accelerometer. Data items included 7 symptoms (Visual Analog Scales) and perceived demands in the preceding period (Likert scales). Activity was measured using mean acceleration in the 3-hour periods preceding and following app data entry. Analysis used within-person correlations of symptoms pairs and both pooled and individual symptom networks derived from graphical vector autoregression. App data was suitable for analysis from 74 participants (90%) comprising 4022 entries representing 77.6% of possible entries. Symptoms varied substantially within individuals and were only weakly autocorrelated. The strongest between-subject symptom correlations were of fatigue with pain (partial coefficient 0.5) and cognitive difficulty with light-headedness (0.41). Pooled within-subject correlations showed fatigue correlated with cognitive difficulty (partial coefficient 0.2) pain (0.19) breathlessness (0.15) and light-headedness (0.12) but not anxiety. Cognitive difficulty was correlated with anxiety and light-headedness (partial coefficients 0.16 and 0.17). Individual participant correlation heatmaps and symptom networks showed no clear patterns indicative of distinct phenotypes. Symptoms, including fatigue, were inconsistently correlated with prior or subsequent physical activity: this may reflect adjustment of activity in response to symptoms. Delayed worsening of symptoms after the highest activity peak was observed in 7 participants.

University of Sheffield: DOI 10.15131/shef.data.
21357762".

**Funding:** The authors received no specific funding
for this work.

**Competing interests:** The authors have declared
that no competing interests exist.

## Conclusion

Symptoms of Long Covid vary within individuals over short time scales, with heterogenous patterns of symptom correlation. The findings are compatible with altered central symptom processing as an additional factor in Long Covid.

## Introduction

Long covid (also known as post COVID-19 condition and post-acute sequelae of covid-19) is a heterogeneous illness which follows acute infection with the SARS-COV-2 virus [1–4]. Its prevalence is currently uncertain because studies have used different criteria for symptoms and time points [5], however recent UK data indicate that 1.2M people (1.9% of the population) have persistent symptoms more than 12 weeks after acute Covid-19 infection and that of these approximately 20% have symptoms which substantially reduce their ability to undertake day-to-day activities [6]. Common symptoms of Long Covid include fatigue, cognitive dysfunction, and breathlessness, but a wide range of other symptoms are commonly present and may predominate in some patients [1, 2, 4]. A characteristic aspect of many patients' experience of Long Covid is the marked and often unpredictable variation of symptoms that occurs over periods of hours and days and which adds to the work of self-management [7, 8] and which is difficult to capture in studies which sample symptoms daily or less frequently [9].

Evidence exists for multiple pathophysiologic mechanisms in Long Covid [3, 5]. Including organ damage [10], persistent changes in inflammatory [11, 12], vascular [13], thrombotic [14], and metabolic processes [15] and autonomic nervous system dysfunction [16, 17]. While some patients show evidence of chronic respiratory disease [18], features of dysfunctional breathing without current respiratory disease have also been observed in Long Covid [19, 20]. Neurological features are well recognised after Covid19. These include changes in grey matter [21] altered smell and taste and cognitive difficulties [22].

A neurological process which may possibly play a role in Long Covid but which has not been widely considered to date, is interoception—the neurological and nonconscious process of sensing, interpreting and regulating the body [23]. Interoception plays an important role in symptom processing, particularly in current models such as embodied predictive interoceptive coding (EPIC) [24, 25] (See S1 Text for more information). In Long Covid, it is plausible that interoception and symptom processing may be altered through changes to sensory afferents (including within the vagus nerve) [26, 27], or in brain areas involved in interoception through damage, inflammation or micro-circulatory changes [28, 29]. If interoceptive pathways are impaired, then the signal to noise ratio of bottom-up signals from the body would be reduced and the interoceptive system would have to work harder to reconcile predictions and signals.

In this study we aimed to examine within- and between- person patterns of symptoms in Long Covid using an intensive longitudinal design. Our objectives were first to quantify the within-person variability of symptoms of Long Covid, second to examine the real-time correlations of different symptoms in the context of daily life, and third to examine the strength of the relationship of symptoms to self-reported demand of activities and objective physical activity. We hypothesised, based on an EPIC model of symptoms, that strong and consistent patterns of association between groups of symptoms or between symptoms and effort and activity would point to accurate interoception of pathophysiological mechanisms impacting on body organs or systems, while weak or inconsistent correlations, would suggest that the pathophysiological processes of Long Covid were being compounded by altered interoception and symptom processing.

## Methods

### Study design

We carried out an intensive longitudinal study (also known as ecological momentary assessment) [30] using self-report data collected through a custom smartphone app supplemented by activity data from a wrist worn accelerometer. The study took place in the UK between July and October 2021 and was delivered remotely. Intensive data collection took place over 14 days. During these days, participants were prompted to enter data 5 times per day (at 3-hour intervals) while wearing the accelerometer continuously. The 14-day intensive data collection period was preceded by a 7-day run-in period during which participants completed the app twice daily. The study design is depicted graphically in Fig 1. The methods were tested in a 7-day pilot study in November 2020- December 2020 with 20 participants to check that the data items had good face validity with participants and were answered appropriately. Ethical approval for the study was granted by Sheffield Hallam University Research Ethics Committee (reference number: ER27968999).

### Patient and public involvement

The project was conceived during discussions with people with Long Covid in August 2020, and the concept refined during further discussions in autumn 2020. Minor changes were made to the wording of the questions in the app in response to feedback from the pilot study.

### Participant inclusion and exclusion criteria

Inclusion criteria were the presence of ongoing physical symptoms which the individual attributed to Long Covid and which followed (by at least 3 months) a recognisable acute infection during the Covid-19 pandemic. These criteria were applied irrespective of whether they had undertaken a PCR test for SARS-CoV-2 or what the result of any test was. The study was restricted to UK residents but not to any geographical area within the UK.

### Recruitment and enrolment

Participants were primarily recruited from the RICOVR [31] database established by Sheffield Hallam University for people living with symptoms of Long Covid. In addition, potential participants were also contacted through the patient-led 'Covid-19 Research Involvement Group'

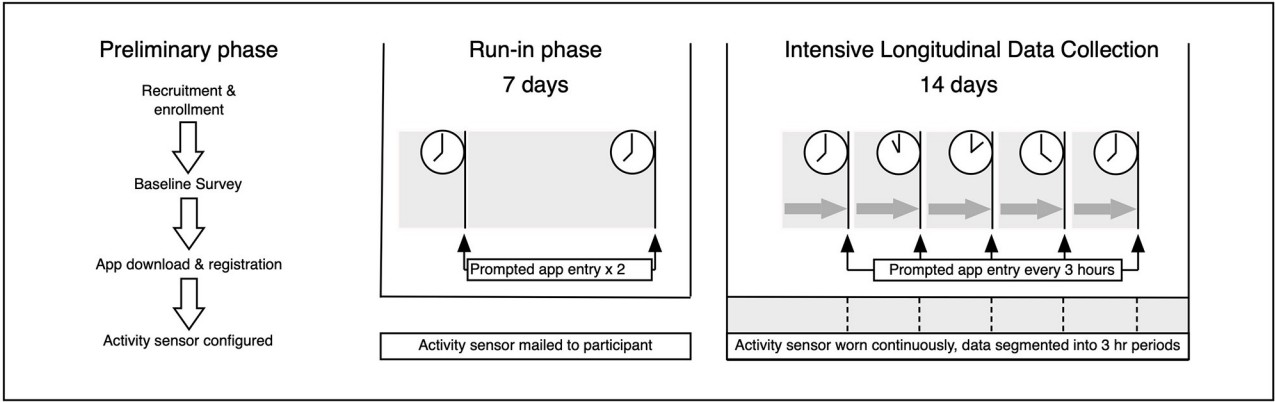

**Fig 1. Schematic representation of the study design.**

on Facebook. Potential participants were sent an invitation email containing a participant information sheet and consent form.

Enrolment was carried out remotely by email and online survey with informed consent confirmed by email. Following consent, individuals were sent a link to the online baseline survey. Eligibility was checked against the survey responses and all participants who met the eligibility criteria were then sent details of how to download and use the app, sent an activity monitor by post, and given the dates to start the run-in week and the intensive two-week data collection with accelerometer wear.

## Baseline survey

The baseline survey included items relating to demographics (age, sex, ethnic group, education, occupation, and socio-economic status) and acute covid illness (month of onset, illness features, testing for SARS-CoV-2, and whether hospitalised). The survey included the following questionnaires to describe symptoms, quality of life and social participation: EQ5D-5L as a measure of health-related quality of life; FACIT-Fatigue, a 13-item measure of fatigue [32], originally developed for cancer but also used in Long Covid, the PROMIS P8a measure of Social Participation [33], the 5 item Post Exertional Malaise questionnaire [34] (modified to refer to the most recent two weeks rather than the originally specified 6 months) and the Interoceptive Accuracy Scale [35] a recently developed self-report measure.

## Smartphone app

A smartphone app was custom-built for the project at the Advanced Wellbeing Research Centre, Sheffield Hallam University. It was made available on both iOS and Android platforms and a link to download it was provided after study enrolment. Data was regularly uploaded from the app to a secure server at Sheffield Hallam University.

The app was designed to send an audible reminder up to three times at pre-specified times during the day (08.00, 11.00, 14.00, 17.00 and 20.00). Data entry was by touchscreen and involved a mix of Likert and visual analogue scale (VAS). Items were framed as either in the present (e.g. "how well do you feel just now?", in the recent past ("since your last data entry. . ."), or in the immediate future ("thinking about the next few hours. . ."). Questions about symptoms were presented as VAS and referred to overall unwellness (reversing the "how well do you feel?" item), fatigue, breathlessness, pain, altered taste or smell, light-headedness or unsteadiness, cognitive difficulties ("difficulty thinking clearly") and feeling anxious or worried. Other questions included how demanding the last few hours had been physically, mentally, and emotionally (each a 4-point Likert scale between not at all and very). Likert scales for demand were converted to numeric scales (range 0–3) and summed to give a measure of combined demand. Data entered into the app was not accessible to the participant but was automatically uploaded at regular intervals to the study database. The wording of items in the app, including additional items not reported here is listed in S1 Table).

The intensive data collection period of 14 days was preceded by a run-in period of 7 days with twice daily data entry in order to familiarise participants with the app and also for the activity sensor to be posted out to them ready for the start of the 14-day period.

## Activity sensor

Participants wore an Axivity AX3 triaxial accelerometer on their non-dominant wrist and were encouraged to wear it at all times except when bathing. Devices were calibrated prior to use and set to start automatically at the beginning of the scheduled 14-day period. Data was

collected at 100Hz and stored in epochs of 5 seconds. At the end of data collection, participants posted the sensor back to the research centre.

## Sample size and missing data

We aimed for a sample size of 80 participants assuming that we would obtain approximately 50 out of 70 possible app entries over 14 days giving a total of 4000 data points. This is equivalent to the largest studies in a recent review of intensive longitudinal studies of symptoms [36] While methods of calculating sample size and power for intensive longitudinal are available [37] they relate only to measurement of fixed effects across participants which was not the focus of this study. Where data points were missing, we did not attempt to interpolate, because we were specifically focusing on within-individual variation, however we specified that data with less than 35 app entries (50% of the possible maximum) would be discarded.

## Data processing

App data were uploaded in real-time from the participant's smartphone and stored securely for subsequent analysis. Data from the activity sensor was aggregated into 30 second epochs. Acceleration was estimated using the Euclidian norm minus one (ENMO) algorithm [38] in the *GGIR* package for R [39] and reported as the mean in milligravitational units. Reference values for this device with the ENMO algorithm are available from the UK biobank population in whom mean acceleration in the period 08.00–20.00 for adults aged 45–64 was between 40 and 50 milligravitational units [40]. We defined broad categories of activity based on published data as sedentary <30; light, 30–100, moderate, 101–400; and vigorous, >400 [41]. We grouped activity data into 3-hour periods immediately preceding scheduled app data entry times and employed two measures of activity: mean activity and the proportion of time spent in different activity categories.

## Overview of analysis

From our analysis of the pilot data we expected that for at least some participants several symptoms would be correlated with each other (this was particularly the case for fatigue). Because of this we chose to analyse both unadjusted correlations (as these reflect the person's experience of the relationship between symptoms) and partial correlations (adjusted for other variables). Analyses considered data at two levels: within-person and between-person. We carried out analyses of symptom variability, correlation, and association with demand at the within-person level. The summary values of these analyses were then combined in a between-person analysis. Additionally we carried out multilevel analysis of symptom networks using graphical vector autoregression which is able to partition data between common networks and individual networks [42].

## Within-person variability of symptoms

Symptom data for the 14-day data collection period were summarised at the individual participant level as mean, range, standard deviation, and root mean square of successive differences (RMSSD). Together these provide measures of symptom magnitude (mean), dispersion (standard deviation) and instability (RMSSD) [43].

## Correlations between symptoms

We estimated the correlation between symptoms in two ways: first by calculating contemporaneous correlations for each symptom pair within each individual and second by using

graphical vector autoregression to study networks of symptoms within and between individuals [44]. Contemporaneous correlations between symptoms used non-parametric (spearman) correlation. This provides a relatively crude measure of association which does not take account of either autocorrelation or collinearity, but which represents the experience of the patient in real time. In order to restrict the reporting of correlations within an individual to symptoms which were commonly present we limited the calculation of correlation coefficients to symptom pairs where the median rating of each symptom was greater than zero. P-values were calculated for correlation coefficients and set at two thresholds: uncorrected (p <0.05) and with Bonferroni correction (p <0.0018).

## Between and within individual networks of symptoms

Symptom networks were derived using Graphical vector autoregression. This technique permits time series data from multiple individuals to be decomposed into between-person and within-person effects. Between-person effects can be understood as the association of a pair of variables based on the average score of each participant. They can be interpreted as indicating that participants who experienced more of one variable also experienced more of another.

Within-person effects can be understood as the association of a pair of variables at each time point within the same individual. Finally within person effects can either be contemporaneous (examining effects at the same time points) or directed (examining the association of one symptom at one time point and the other at the next). Contemporaneous associations can be interpreted as indicating that when participants experienced more of one variable, they also experienced more of another. Directed associations can be interpreted as indicating that when participants experienced more of one variable they then experienced more of the same or another variable at the next time point.

Analysis used joint multivariate least absolute shrinkage and selection operator (LASSO) estimation [42]. Optimal tuning parameters were selected using the Extended Bayesian Information Criterion. Data were analysed as a time series in order to examine both contemporaneous and directed (temporal) effects. Lagged values were not used for the first entry of each day or after missing data points and no missing data was imputed. Results were reported as partial correlations in order to account for the effects of other variables.

## Relationship between symptoms and physical activity

The relationship between effort and symptoms was examined in two ways. First, we examined the within person correlations of fatigue and overall unwellness with data from the accelerometer and subjective demand over the preceding 3 hours. Second, we looked for post exertional symptom exacerbation by converting data points for fatigue and overall unwellness into normalised (z-) scores at the individual level, calculating an unweighted moving average (over 4 points) and plotting values for the 72 hours before and after the most active 3-hour period during the 14 days. We defined post-peak exacerbation of symptoms as one or more moving average z-scores of $\geq 1.3$ (equivalent to the $90^{th}$ centile of a normal distribution) for either fatigue or overall unwellness, which occurred between 12 and 60 hours after the peak activity period. We repeated a similar process for the 72 hours before and after the 3-hour period with the highest combined subjective burden (physical, cognitive, and emotional). Where the highest burden occurred in more than one period, we selected the high-burden period with the highest objective physical activity score.

### Relationship of findings to baseline data

All analyses were conducted independently of participants' baseline characteristics. Once analysis was complete, we examined the distribution of various measures in three groups: those with a positive PCR test, those with a negative PCR test and those with no history of testing. As almost all those with no history of testing had their acute illness prior to the widespread availability of PCR testing, we compared those with a negative test against all others by t-test.

All analyses were conducted in R 4.0 (R foundation, Vienna) with the *graphicalVAR* package for R [45].

## Results

### Participants and data

**Completeness of data.**   Between July and September 2021, 82 eligible individuals consented to take part in the study and were registered to use the app. Data was suitable for analysis from 74 (90%). Of the remainder, 7 completed less than half the possible data entries and one reported that they had worked a period of night shift during the data collection period.

App data from the 74 included participants comprised a total of 4022 entries collected on 1025 person-days representing 78% of possible entries. The median number of completed entries per participant was 55.5 (IQR 46 to 61) with data entry on all 14 days by 66, on 13 days by 7, and on 10 days by 1. S2 Table shows the distribution of completed entries by participant and time of day. There was no association between number of completed entries and mean self-reported fatigue (spearman's rho -0.05). Data was entered within 10 minutes of the first entry prompt in 54.2% of instances, between 11 and 35 minutes in 32.7% instances and more than 35 minutes after the first prompt in 13%. Data entries were more likely to be delayed or omitted for the first (08.00) prompt of the day compared to others (S3 Table).

Accelerometer data was available for 69 (93.2%) of participants with sufficient app data. Of the remaining 5, three returned the accelerometer with data which could not be analysed, one only used the accelerometer for a few days and one device was lost in transit.

**Participant characteristics.**   Participants were aged between 21 and 64; the median age was 50, (IQR = 42 to 54). 63 (85.1%) were female. 67 (90.5%) were of White British ethnicity, and 46 (62.1%) had been educated to university degree level. 42 (56.8%) were currently in work or equivalent activity with 22 (29.7%) reporting inability to work because of Long Covid.

The duration of symptoms following participants' acute covid illness was < 6 months for 3 participants, 6–12 months for 8, 12–18 months for 47 and >18 months for 5 (duration of illness data was not reported for 11). 8 participants had been hospitalised with their acute covid illness, though none had required invasive ventilation. 54 (73%) participants described having a PCR test for Covid-19: 27 had been positive and 27 negative. Of the 20 not tested, 17 had their acute illness in the period before widespread testing was available. Almost all participants had their first symptoms before vaccination was widely available and we did not collect additional information on this.

**Baseline survey data.**   At baseline most participants reported substantially impaired quality of life: median EQ-5D-5L index was 0.64 (IQR 0.37 to 0.75) and median EQ5D VAS was 40 (30 to 50). Table 1 describes the baseline survey results. Analysis of variance showed no significant difference in baseline measures between those testing positive for Covid-19, negative or not tested.

**Symptom intensity and variability.**   Fig 2 shows the mean and range for each symptom by individual participant. Fatigue was the most frequently reported symptom: 73 (98.6%) participants reported a VAS score for fatigue ≥15/100 on at least 50% of entries (equivalent to

**Table 1. Summary of baseline survey measures.**

| Measure | Median | IQR | Min | Max |
|---|---|---|---|---|
| Age | 50 | 42, 54 | 21 | 64 |
| Quality of Life (EQ5D) | 0.64 | 0.37,0.75 | -0.18 | 0.95 |
| Quality of Life (EQ-VAS) | 40.00 | 30,50 | 2.00 | 80.00 |
| Fatigue (FACIT-Fatigue 13 item)* | 15.00 | 8, 20 | 2.00 | 32.00 |
| Social adaptation (Promis P8a) | 11.00 | 8,18 | 7.00 | 27.00 |
| Physical Symptoms (PHQ-15) | 15.00 | 13,18 | 5.00 | 29.00 |
| Interoceptive Accuracy (IAS) | 79.00 | 71, 88 | 46.00 | 105.00 |
| Post-exertional malaise scale | 15.00 | 10, 18 | 4.00 | 20.00 |

*FACIT-Fatigue scale is a quality-of-life related scale which is scored such that low values represent greater fatigue and high values represent less fatigue.

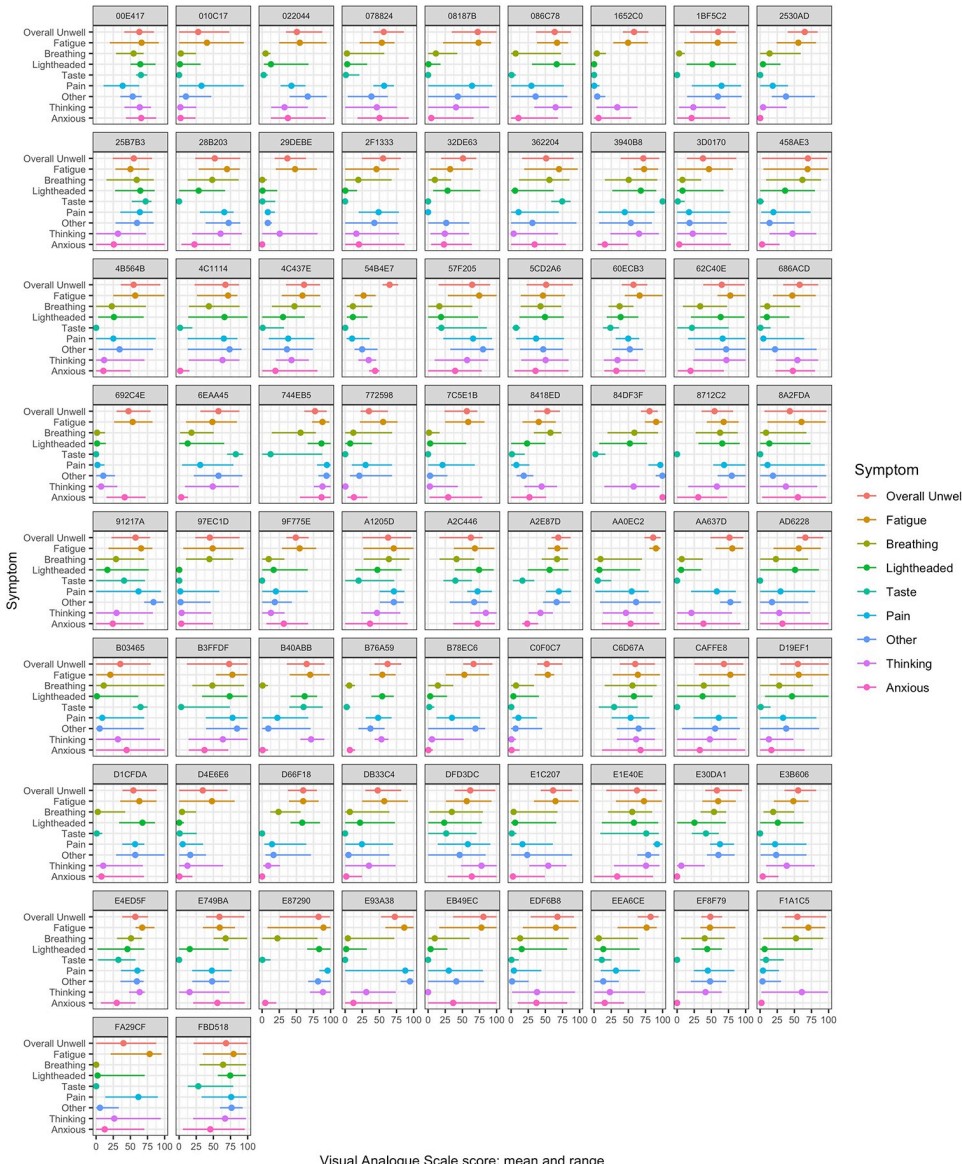

**Fig 2. Mean and range for each symptom by participant ID.** Values represent rating (0–100) on visual analogue scale in the study app.

one point on a 7-item Likert scale). Using this criterion, difficulties with taste or smell were least frequent (27% of participants) while other symptoms occurred in between 51 and 76% of participants. Across all participants, the median of the within-person mean VAS scores for overall unwellness was 59 and for fatigue was 60. Fig 3 shows the standard deviation and

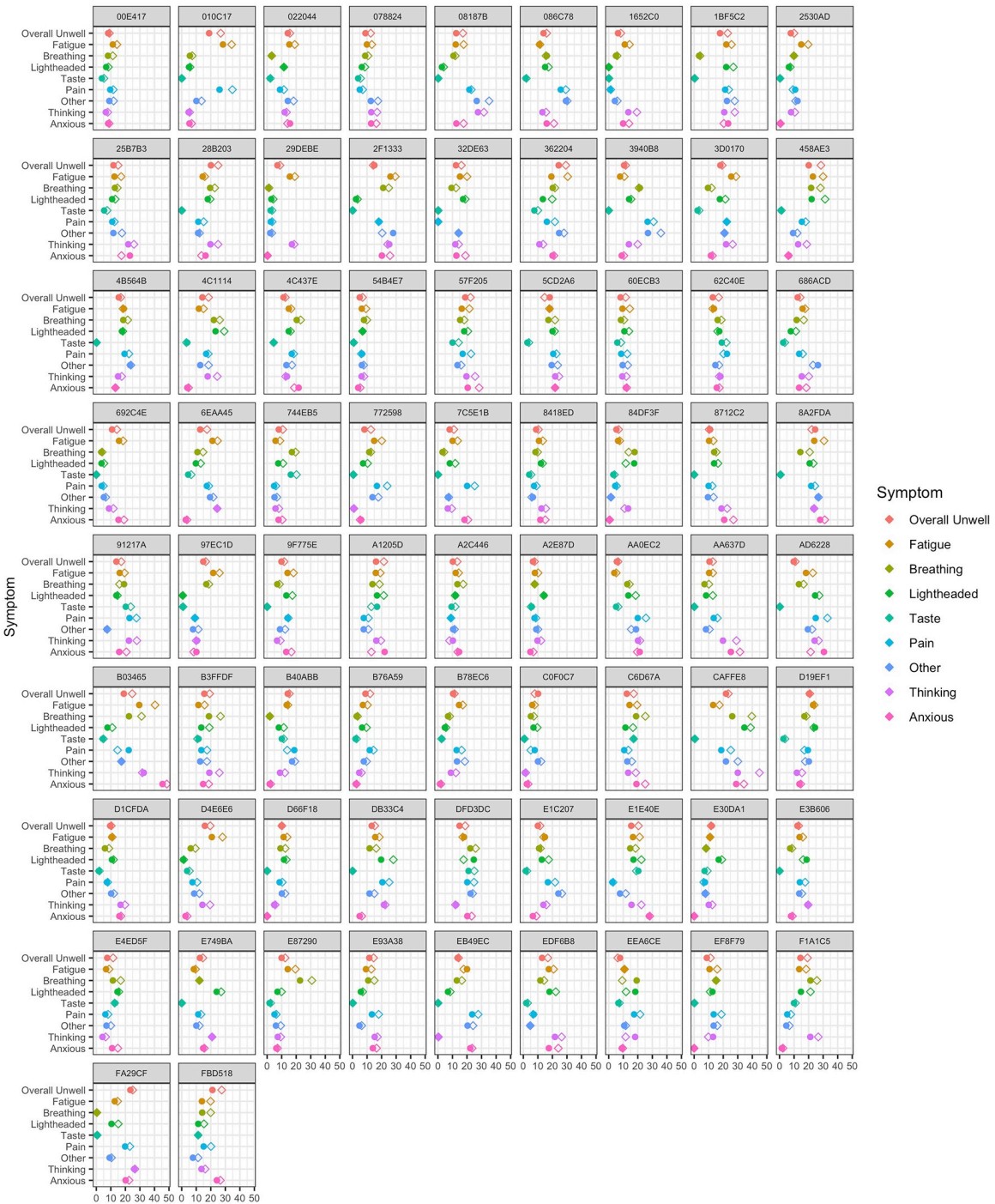

**Fig 3. Standard deviation (circles) and root mean square of successive differences (diamonds) for each symptom by participant ID.** Values represent rating (0–100) on visual analogue scale in the study app.

RMSSD of each symptom by individual participant. Table 2 summarises this data. Despite the substantial within-person variation of symptoms seen in Fig 2, most participants showed relatively small amounts of diurnal variation. Linear regression found that in the case of fatigue, 16 (21.6%) had a statistically significant increase in VAS across the day and 4 (5.4%) had a statistically significant decrease. In the case of overall unwellness, 8 (10.8%) participants had a significant increase across the day and 7 (9.5%) had a significant decrease.

In summary there was substantial variation in symptoms within and between participants. The finding that the RMSSD (a measure of instability within a time series) was greater than the standard deviation for most symptoms in most participants, indicates that symptom intensity at one time point was relatively independent of symptom intensity at the preceding time point. This is examined further in the symptom networks analysis.

## Correlations between symptoms

**Associations between symptoms: Within person analysis.**   Fig 4 shows the distribution of within-person correlations between symptom pairs. The figure includes the number of correlations reported for each symptom pair which varies because if either symptom had a median VAS <15 no correlation was calculated. The strongest mean correlation was between fatigue and overall unwellness (mean rho = 0.57). At the individual participant level, this correlation was statistically significant after Bonferroni correction (p-value < .0018) for all but 6 individuals. Overall, 285/1403 (20.3%) individual associations between features showed strong correlations ($\geq$0.5), 394 (28.1%) were moderate (0.3–4.9), and 378 (26.9%) were weak (0.1 to 0.29). 87 (6.2%) correlations were negative, of which 79 were weak and 8 were moderate; 259 (18.5%) showed no correlation (-0.09 to 0.09). For most symptoms there was no relationship between the correlation coefficient of symptom pairs and their mean VAS scores. The exception to this was light-headedness: higher VAS levels were associated with stronger correlations with other symptoms, particularly fatigue (p = .0004) and overall unwellness (p = .003) (S1 Fig).

Heatmaps of unadjusted correlations for each individual are shown in Fig 5 and S2 Fig. Fig 5 shows data from the 39 participants for whom the median VAS was greater than zero for all symptoms, while S2 Fig shows data from all participants.

**Networks of symptoms.**   Fig 6 summarises the findings of the network analysis of symptoms in which relationships between variables are reported as partial correlations.

**Table 2. Summary of symptom self-report values per participant.**

| Symptom | Above threshold | | Mean of VAS | | | | SD of VAS | | RMSSD of VAS | |
|---|---|---|---|---|---|---|---|---|---|---|
| | N | (%) | Median | IQR | Min | Max | Median | IQR | Median | RM.IQR |
| Overall | 74 | 100.0 | 59 | 52,66 | 28 | 86 | 12 | 10,15 | 14 | 11,19 |
| Fatigue | 73 | 98.6 | 60 | 53,71 | 21 | 91 | 14 | 11,17 | 17 | 13,20 |
| Breathing | 38 | 51.4 | 19 | 7,49 | 0 | 68 | 12 | 8,17 | 15 | 10,19 |
| Lightheaded | 39 | 52.7 | 23 | 6,54 | 0 | 87 | 13 | 7,17 | 14 | 11,19 |
| Taste | 20 | 27.0 | 1 | 0,19 | 0 | 100 | 2 | 0,6 | 3 | 0,7 |
| Pain | 56 | 75.7 | 38 | 19,62 | 0 | 97 | 13 | 8,19 | 16 | 9,22 |
| Thinking | 55 | 74.3 | 38 | 18,57 | 0 | 89 | 14 | 10,20 | 17 | 12,24 |
| Anxious | 41 | 55.4 | 23 | 4,37 | 0 | 100 | 13 | 6,20 | 15 | 7,21 |

*Note*. Threshold refers to participants whose median VAS score for each feature was $\geq$15 (equivalent to one point on a 7-item Likert scale). VAS = visual analogue scale; SD = standard deviation; RMSSD = root mean square of successive differences, IQR = interquartile range. Mean, SD and RMSSD refer to within-person distributions, median and IQR refer to between-person distribution of within-person mean etc. Overall refers to a single summary 'how unwell do you feel now?' question

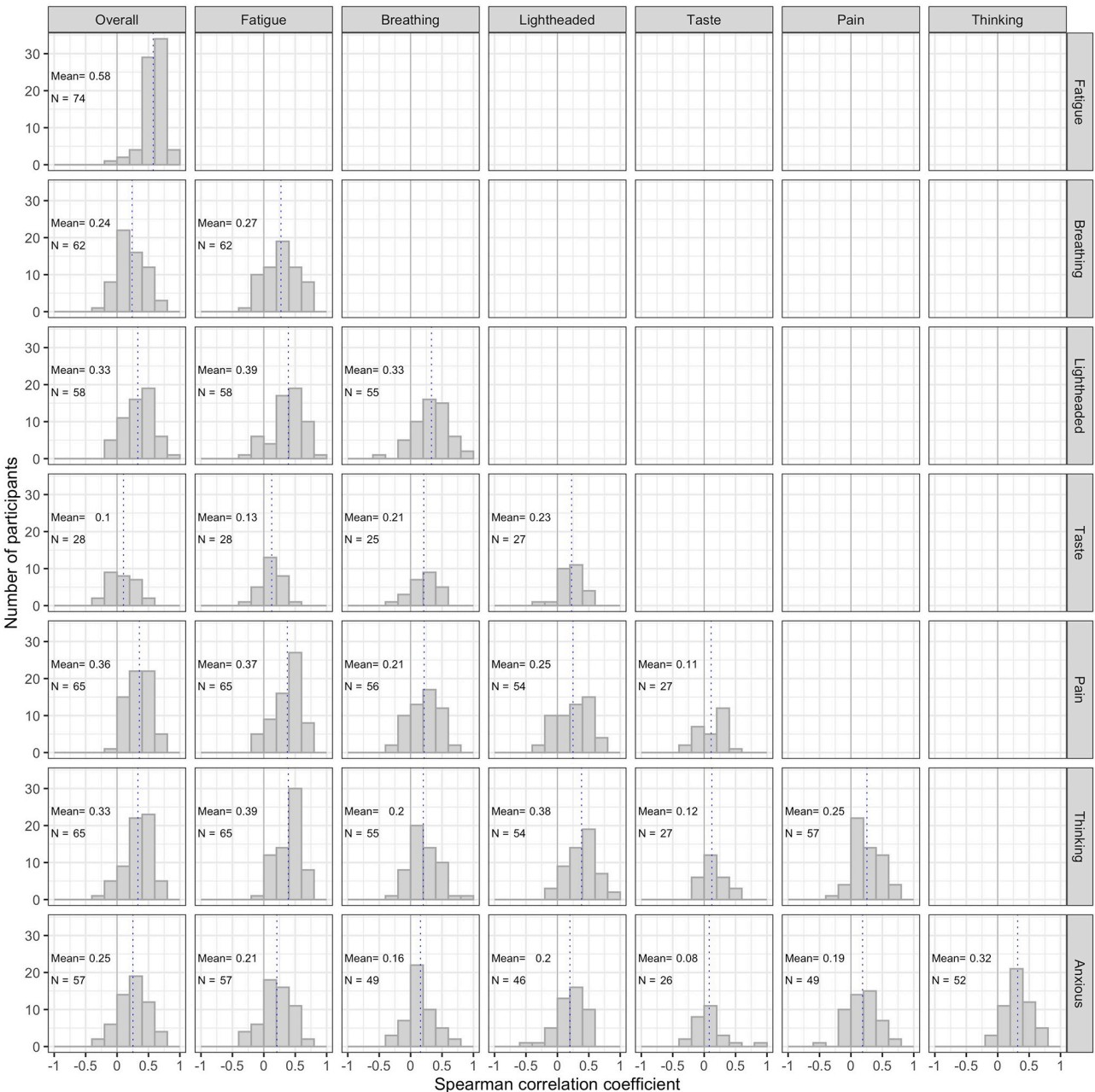

**Fig 4. Within-person associations between symptom pairs.** Solid line indicates zero, dotted line indicates mean of correlations. Correlations only reported for individuals where both symptoms had a median value greater than zero. Labels in each sub-figure indicate number of individual correlations reported and mean correlation coefficient.

**Pooled between-person analysis.** The partial correlations represent the relationships between the mean VAS per person. The strongest associations were of fatigue with pain (partial correlation coefficient = 0.54), and light-headedness with cognitive difficulty and breathlessness (0.42 and 0.32 respectively).

**Pooled within-person contemporaneous analysis.** The pooled within-subject contemporaneous correlations were less strong than the between-subject ones. Because these partial correlations account for other variables, they are also smaller than the unadjusted correlations

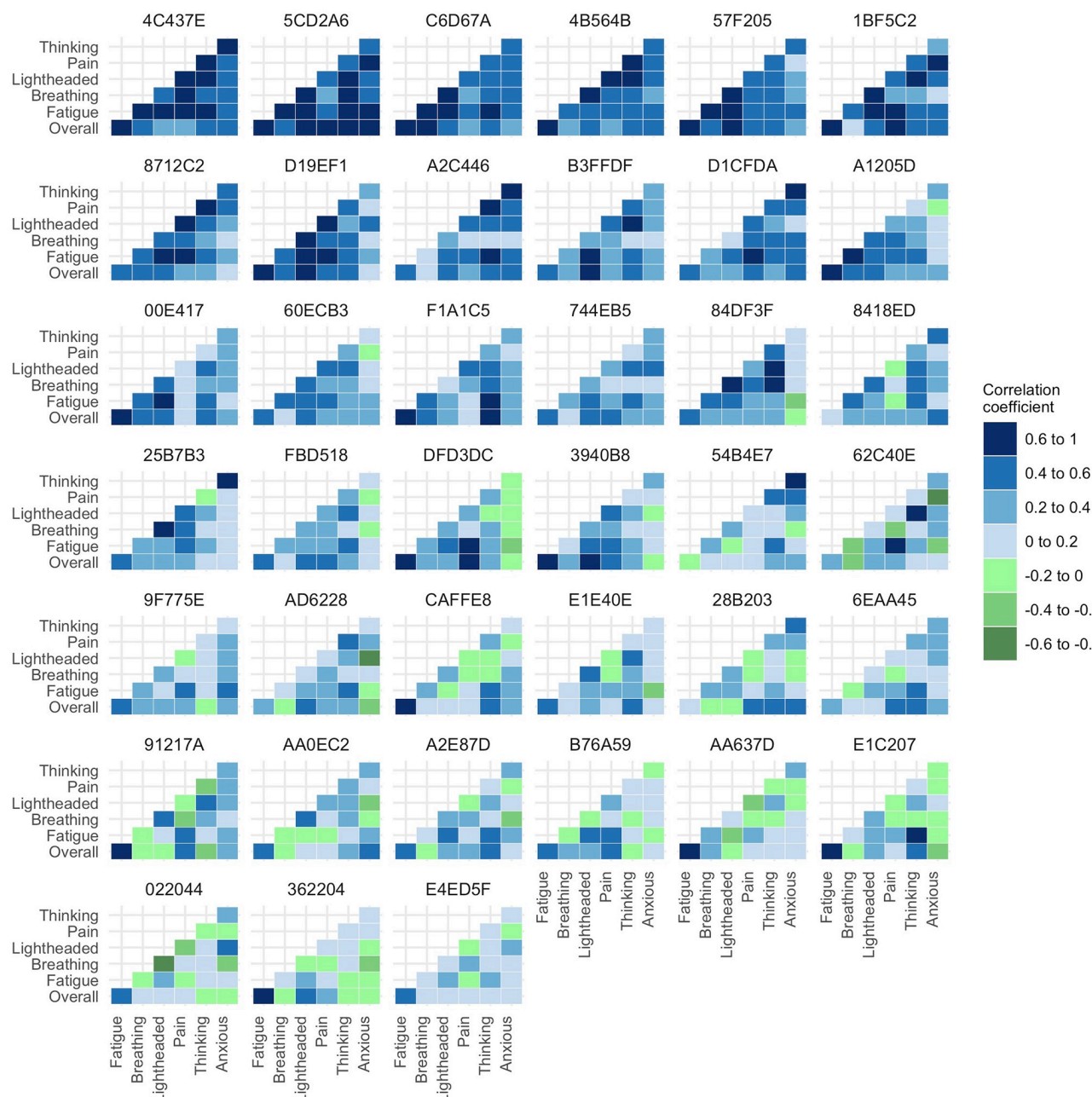

**Fig 5. Heatmap of unadjusted correlations between symptoms at the individual participant level.** Figure shows only those participants who experienced all relevant symptoms at sufficient level to be included in the correlation analysis.

shown in Fig 5 (e.g. for fatigue and cognitive difficulty partial correlation = 0.21, unadjusted correlation = 0.39; for light-headedness and breathlessness partial correlation = 0.16, unadjusted correlation = 0.33). Fatigue was positively correlated with pain, light-headedness, breathlessness, and cognitive difficulties but not with anxiety. Activity in the preceding 3 hours was weakly correlated with breathlessness (0.12) but not correlated with fatigue (-0.04).

**Pooled within-person temporal analysis.** The pooled within subject temporal network analysis of symptoms showed moderate autocorrelation (indicated by an arrow looping back

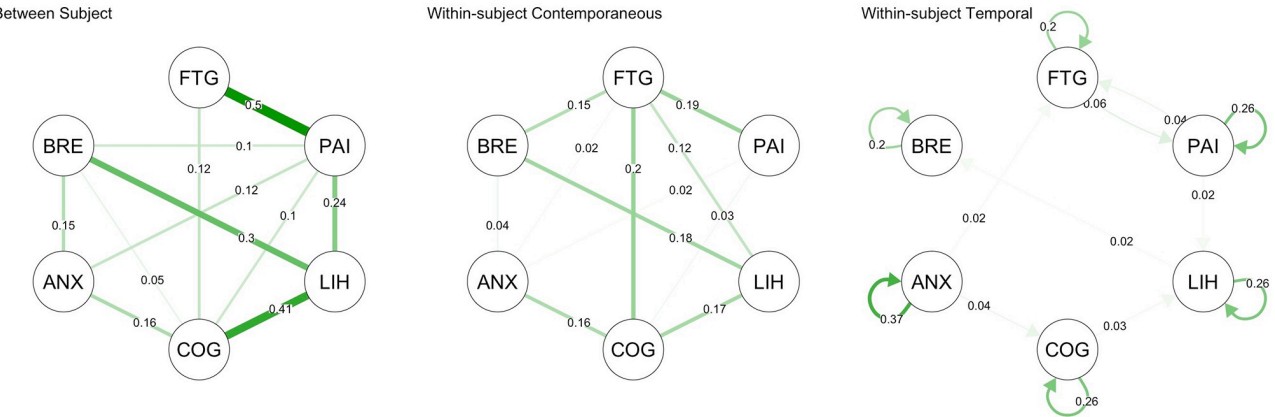

**Fig 6. Summary network of symptoms.** Between-subject network shows partial correlations based on individual participant mean values, Within-subject contemporaneous network maps statistically significant partial correlations of data points at the same time point; Within-subject Temporal network shows statistically significant partial correlations at lag(1) and autocorrelations. FTG, fatigue; PAI, pain; LIH, light-headedness; COG, cognitive difficulty; ANX, anxiety. BRE, breathlessness.

to the same symptom) for anxiety (coefficient = 0.36) with weak autocorrelation for other symptoms (between 0.2 and 0.27) and no significant autocorrelation for activity. There were several very weak lagged partial correlations between symptoms but all had values <0.04.

**Individual within-person networks.** Figs 7 and 8 show the contemporaneous and directed (temporal) correlation networks for each individual participant. In both these plots there are only a few commonly occurring patterns. Statistically significant partial correlations were detected between fatigue and cognitive difficulties in 24 (32.4%), between breathlessness and light-headedness in 17 participants (23.0%),between cognitive difficulties and anxiety in 17 (23.0%) and between fatigue and light-headedness in 13 (17.6%). Directed (temporal) correlation networks varied markedly between individuals with most having no significant directed correlations but 6 patients showing complex networks involving multiple statistically significant associations between symptoms (e.g. patient ID 84DF3F). While the pooled data showed moderate levels of autocorrelation for all symptoms, the individual analysis found that 26 (35.2%) participants had no statistically significant autocorrelations, 24 (32.4%) had one, 16 (21.6%) had two, 7 (9.4%) had three and 1 participant had four.

## Analysis of physical activity and association with symptoms

**Physical activity data.** Physical activity data was available for 69 of the 74 participants included in the app data analysis. The findings are summarised in Table 3 which describes the distribution of summary measures of physical activity per individual.

The low mean activity and low proportion of daytime spent non-sedentary (median = 27%) indicates that most participants had low overall activity. For periods of moderate or vigorous activity lasting at least 10 minutes, 10 (14.5%) participants averaged more than 150 minutes per week (meeting WHO recommended levels), 15 (21.7%) averaged between 60 and 150 minutes, 23 (33.3%) averaged less than 60 minutes and 21 (30.4%) had no periods of sustained moderate or vigorous activity. For any moderate to vigorous physical activity regardless of duration, 17 (24.6%) participants averaged more than 60 minutes per day, 21 (30.4%) averaged 30–59 minutes, 26 (37.7%) averaged 10–29 minutes and 5 (7.2%) averaged less than 10 minutes per day.

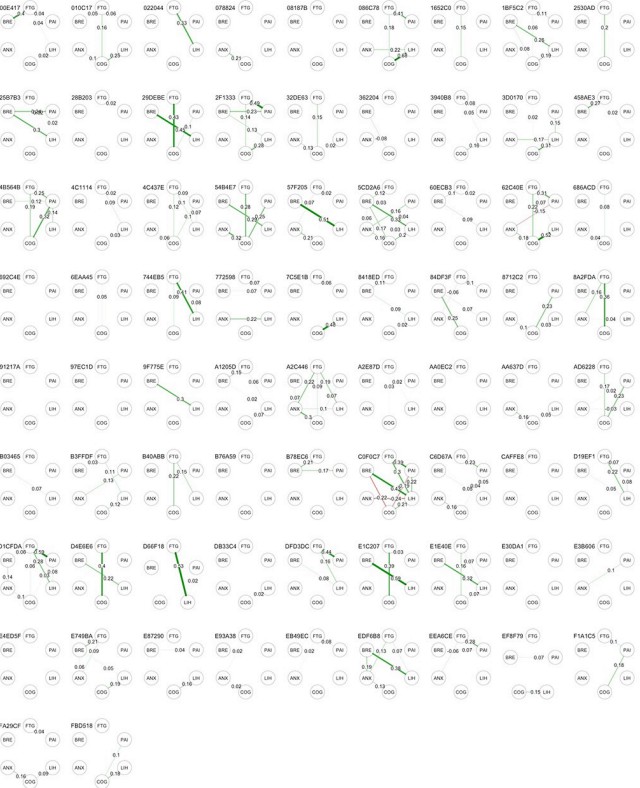

**Fig 7. Within-subject contemporaneous symptom networks.** Figures show statistically significant partial correlations of data points at the same time point. FTG, fatigue; PAI, pain; LIH, light-headedness; COG, cognitive difficulty; ANX, anxiety. BRE, breathlessness.

**Correlations between objective and subjective assessments of activity.** Fig 9 summarises the within-person correlations between objective physical activity, recalled subjective demand and current fatigue. Correlations between subjective physical demand and objective physical activity were moderate to strong (mean correlation 0.31). In contrast, correlations between recent activity and fatigue were weak or absent (mean correlation -0.09 for activity in the preceding 3 hours and -0.02 for activity in the preceding one hour. Fatigue was more strongly correlated with subjective demand (mean correlation 0.16) than objectively measured activity. Physical activity in the 3 hours following an app data entry was negatively correlated with current fatigue (mean correlation -0.11). Similar findings were observed for overall unwellness (S3 Fig).

**Changes in symptoms following peak activity.** Fig 10 shows data for overall unwellness in the 72 hours before and after the 3-hour period of peak physical activity as a moving average (over 4 readings). Data from 7 (10.1%) individuals met our criteria for post-exertional symptom exacerbation: their data are shown as bold lines. Data from the remaining individuals are shown as faint lines.

**Relationship of findings to PCR status.** Fig 11 shows the distributions of selected important variables from the baseline survey, accelerometery and self-report data in relation to PCR status. The only significant difference was in altered taste (which was less common in those with negative PCR than others t = 3.2, p = 0.002).

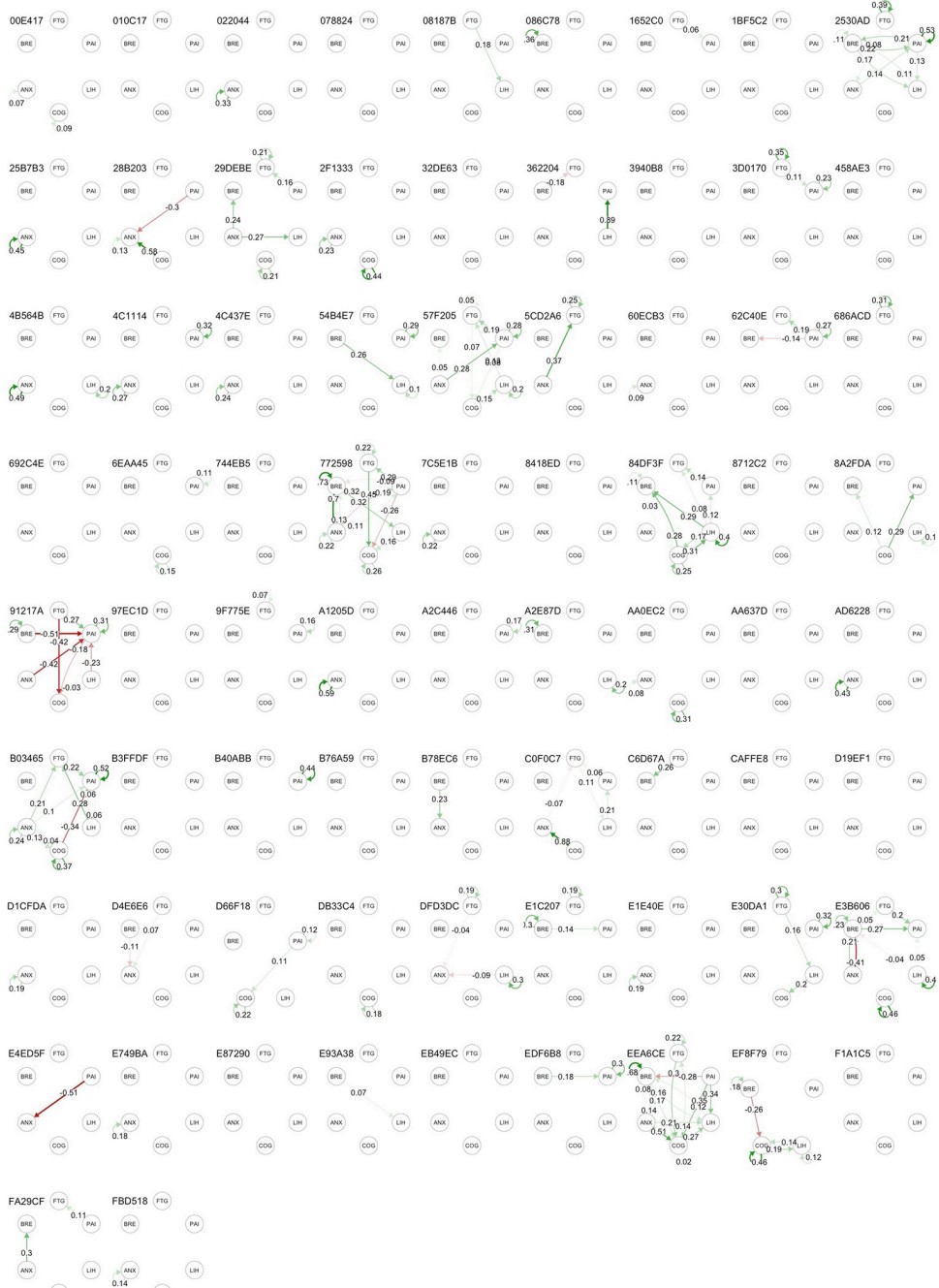

**Fig 8. Within-subject temporal (lagged) symptom networks.** Figures show statistically significant partial correlations of data between consecutive time point. FTG, fatigue; PAI, pain; LIH, light-headedness; COG, cognitive difficulty; ANX, anxiety. BRE, breathlessness.

## Discussion

### Summary of main findings

This intensive longitudinal study of symptoms and physical activity in Long Covid found marked within-person variation in symptoms and between-person differences in the patterns

**Table 3. Distribution of summary measures of physical activity data at individual participant level.**

| Measure | Median | IQR |
|---|---|---|
| Mean Activity (08.00–20:00) | 30.0 | 24.3, 36.7 |
| SD of activity | 33.9 | 28.0, 40.2 |
| Proportion of time non-sedentary | 0.35 | 0.29, 0.41 |
| Minutes per day moderate or vigorous physical activity *(any duration)* | 34 | 18, 58 |
| Minutes per week moderate or vigorous physical activity *(at least 10 consecutive minutes)* | 16 | 0, 92 |

*Note*. Data relates to physical activity between 08:00 and 20:00. Mean and SD activity is expressed in milligravitational units where values <30 are deemed sedentary and values >100 indicate moderate activity. Mean activity refers to the overall mean across the 14 days. Standard deviation refers to the mean of the 3 hourly estimates of standard deviation. Hours per week in moderate or vigorous activity refers to time in continuous blocks of 10 minutes so does not include all moderate or vigorous activity.

of correlation between symptoms. Concurrent associations between symptoms and measured physical activity were weak although a delayed increase in symptoms following peak activity was observed in a small number of participants. The findings quantify the experience of unpredictability reported by patients in accounts of the difficulties of living with Long Covid [7, 8].

## Strengths and limitations

The study used well-established methods to obtain data and had excellent completion rates with the app. Self-report using visual analogue scales on a smartphone app gives participants control over the data they enter and reduces risks of bias when completing scales either at a later time or with a researcher [46]. Importantly, there was patient involvement in the design of the app and in the interpretation of the results. Analysis used a combination of idiographic (within individual) and nomothetic (between individuals) methods including state of the art graphical vector autoregression modelling [42, 47]. Completion rates (78% of possible entries) were comparable to the 80% typically expected in intensive longitudinal studies [48].

The main limitation is that the sample was largely white, female, middle-aged and well-educated. This reflects the opportunistic sample taken from an online panel of research volunteers promoted by peer-support groups, however this has been seen in other studies [8]. This also meant that additional clinical data obtained during routine or specialised care was not available to supplement the data generated in the study. Approximately one third of participants had their initial illness before the widespread availability of PCR testing for SARS-Cov-2 and another third reported that their PCR test had been negative. This raises the possibility that not all participants' symptoms were sequelae of covid infection [49] however we were unable to test serology in this study nor check records for prior symptoms.

Although the number of participants (N = 74) was small compared with large cross-sectional studies or less intensively sampled time series, it is large for an intensive longitudinal study of symptomatic patients: only 1 of 21 studies in a recent review had a larger number of participants and comparable sampling frequency and duration [36]. The time windows used for symptoms and activity (3 hours) were chosen pragmatically but may have been too long to capture meaningful variation. However, they were designed to fit around natural periods in the day and more than 5 entries per day would have added to the burden on participants and increased the likelihood of missing data. While more intensive sampling is sometimes used, it tends to be for shorter periods and we chose to prioritise the number of study days over the intensity of sampling within days.

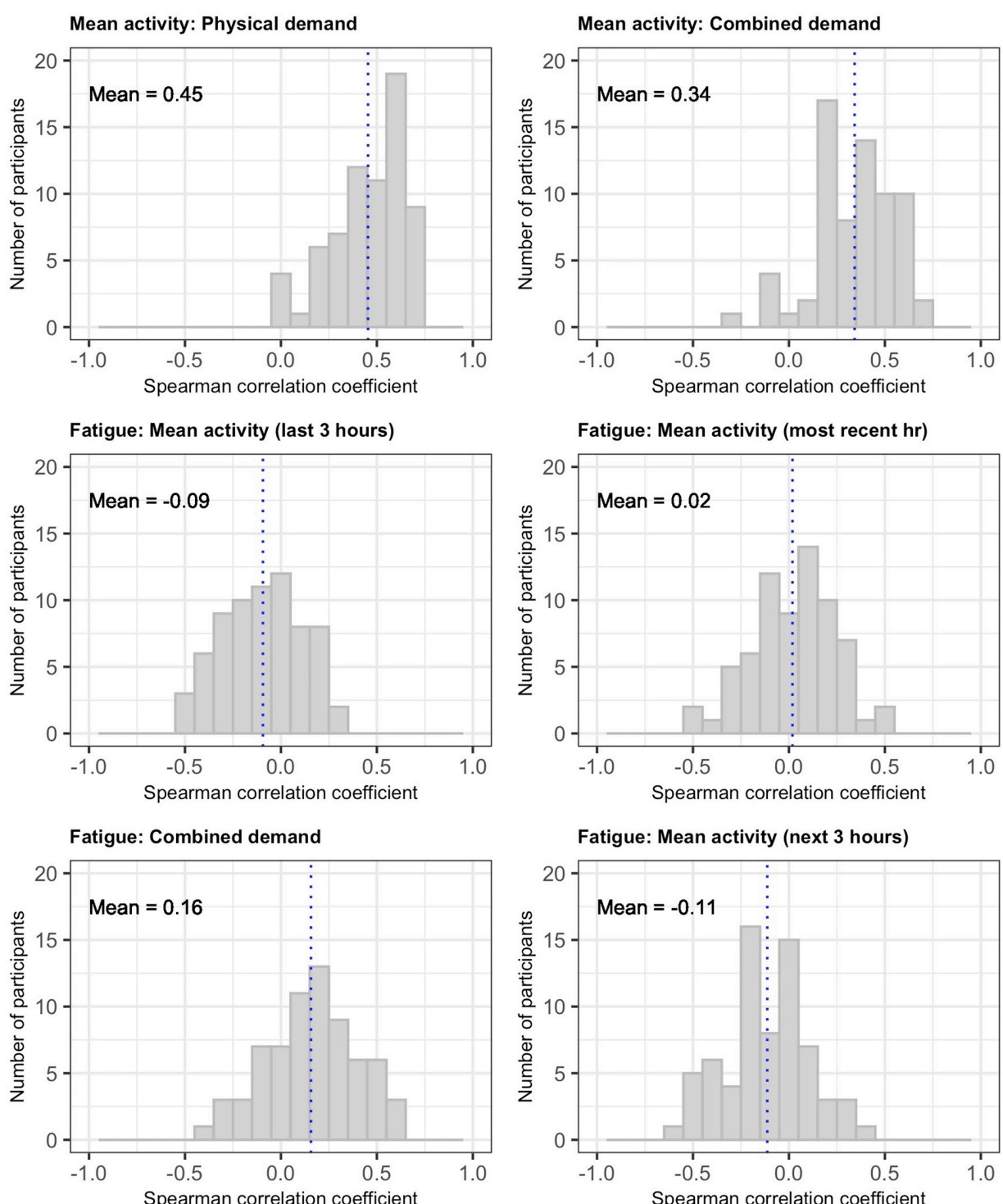

**Fig 9. Distribution of within-person correlations between objective activity from accelerometer and subjective measures of fatigue and demand (physical alone or combined physical, cognitive, and emotional).** All activity measures refer to the 3-hour period before app data entry except where indicated.

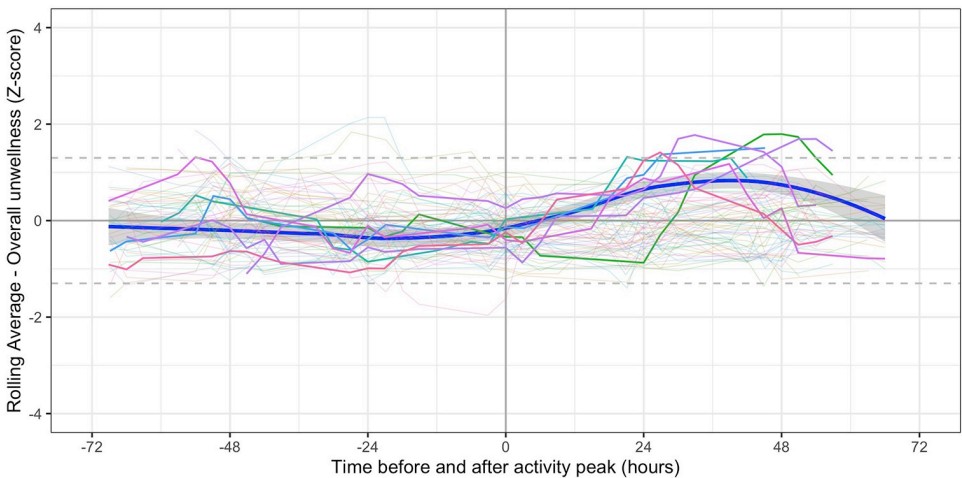

**Fig 10. Symptoms in relation to peak period of physical activity.** Bold lines indicated participants (n = 7) in whom the moving average z-score (for either fatigue or overall unwellness) was ≥1.3 between 12 and 60 hours post peak. Pale lines represent other participants. Smoothed regression line fitted to data from the 7 participants only.

The number of symptoms in the app was deliberately restricted in order to keep the number of variables manageable for both app use and analysis. While the selection covered the main features of Long Covid it did include some symptom groups: pain did not differentiate between body areas and light-headedness and unsteadiness were combined. We did not include specific measures of mood or of cognitive features (such as symptom-focusing) as patient input into the design of the study saw this as potentially implying psychological causation of symptoms.

The observed weak associations between activity and fatigue need to be interpreted with caution for several reasons. First, participants were relatively inactive–only one third managed more than 1 hour of moderate or vigorous physical activity (in periods of at least 10 minutes) per week. This means they may not have been sufficiently active during the 14 days of the study to exacerbate symptoms above existing levels. Second, activity and fatigue are impossible to disentangle; for instance low activity may represent a period of deliberate pacing or recovery to improve symptoms and high activity is more likely to occur during a spell of low symptom burden. Third, weak or absent associations between activity and fatigue have also been observed in intensive longitudinal studies of other conditions including osteoarthritis [50] and multiple sclerosis [51]. Finally, while we observed delayed increase in symptoms following peak activity in a few patients, based on a single peak, we were not able to examine the effects of multiple or overlapping activity peaks.

We were unable to measure interoception directly, and relied only on a single questionnaire [35] to measure a phenomenon about which understanding is currently evolving [52]. In particular the questionnaire does not directly differentiate between belief in the accuracy of interoception and actual accuracy of interoception and so may have missed important differences in interoceptive ability.

## Interpretation

Four aspects of our findings point to a potential role of disordered interoception and symptom processing in Long Covid. These are (1) the within person variability of symptoms; (2) the

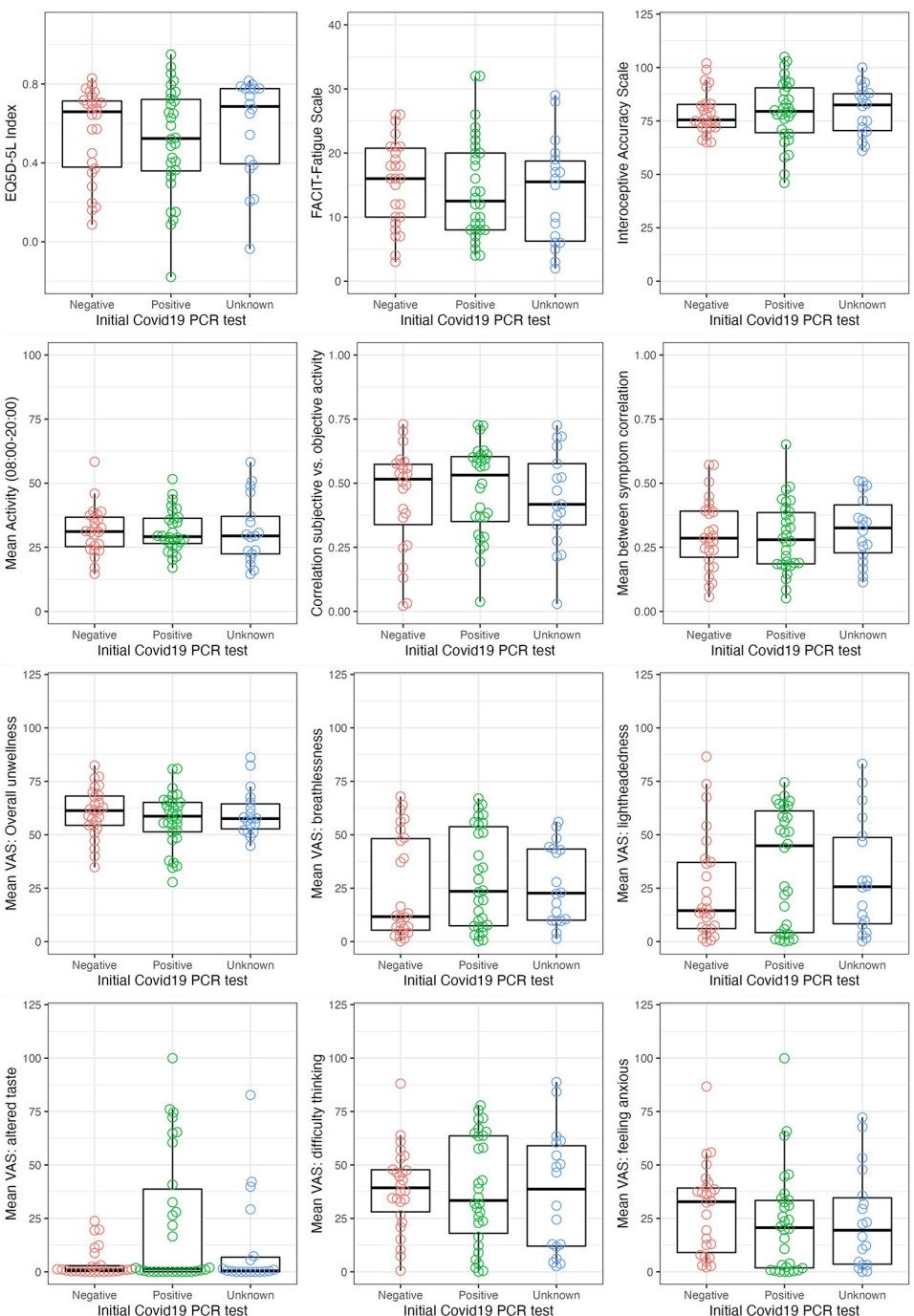

**Fig 11. Distribution of individual participant values by PCR testing status.** Unknown category represents participants who did not have a PCR test during their initial illness. VAS = Visual Analogue Scale.

relationship between symptoms and activity, (3) the between person variation in patterns of symptoms; (4) the cluster of light-headedness, breathlessness, and cognitive difficulty.

The marked within-person variability indicates that symptoms vary within and between days in ways which appear largely unpredictable and inconsistently related to activity or demand. While this variation could be explained by highly varying pathophysiology, it may

also be explained by disturbed processing of interoceptive signals such that symptoms represent inaccurate markers of the body's state. The weak association of symptoms, including fatigue, with activity is similar to that seen in musculoskeletal [50] and neurological disease [51]. This does not invalidate patients' experience of fatigue and other symptoms, rather it emphasises its complexity. In suggesting the possibility that altered interoception may play a role in Long Covid we are not arguing that other mechanisms are not present; rather our data suggests that, on their own, peripheral mechanisms may not fully account for the patterns of symptoms or their severity. Indeed, peripheral pathophysiology and central processes such as interoception are complementary and may be causally linked both through changes in brain function due to vascular and inflammatory processes underpinning changes in interoceptive processing and through the close links between interoception and autonomic signalling [23]. One cluster of associations that did feature in our data was of light-headedness, cognitive difficulty, and breathlessness. These may relate to changes in areas of the brain involved in interoceptive processing [28] or in autonomic regulation [3, 53]. Dysfunctional breathing [19, 54] which might account for some of this cluster can also be understood using an interoception framework [55].

Together these findings can be viewed from the perspective of an embodied predictive interoceptive coding model of symptoms [24, 56]. Indeed, viewing physical symptoms as altered body signals from the body through an impaired interoceptive system may be analogous to the widely recognised parosmia observed during recovery of taste and smell after covid [57]. Addressing Long Covid from this perspective has the potential to be both non-stigmatising and to suggest additional therapeutic approaches and treatments. Several interventions appear capable of improving interoception: these include vagal nerve stimulation and behavioural techniques such as slow paced breathing [58]. Some of these are currently being provided in interventions for Long Covid and all warrant further study. Further research should use methods similar to those described here, with more closely defined groups of patients and in parallel with investigations of other pathophysiological mechanisms or within trials of interventions.

## Conclusion

Symptoms of Long Covid vary within individuals over short time scales, with heterogenous patterns of symptom correlation and weak associations with concurrent or preceding physical activity. These findings are in keeping with an embodied predictive interoceptive coding model of symptoms, suggesting that Long Covid is associated with changes to the way the brain processes signals from the body.

## Supporting information

**S1 Checklist. Adapted STROBE Checklist for Reporting EMA Studies (CREMAS).**
(DOCX)

**S1 Fig. Relationship between correlation coefficient between symptoms and the mean VAS score on which correlation is based.**
(DOCX)

**S2 Fig. Heatmap of unadjusted correlations between symptoms at the individual participant level.**
(DOCX)

**S3 Fig. Distribution of within-person correlations between objective activity from accelerometer and subjective measures of overall unwellness and demand (physical alone or

**combined physical, cognitive and emotional).**
(DOCX)

**S1 Table. Smartphone app questions.**
(DOCX)

**S2 Table. Data entries by time by included participant.**
(DOCX)

**S3 Table. Promptness of data entry by time of day.**
(DOCX)

**S1 Text.**
(DOCX)

## Acknowledgments

We would like to thank the participants who took part in this study and the people with Long Covid who gave advice on the project throughout, including advising on the study design, interpreting the findings, and critically reading the manuscript.

For the purpose of open access, the author has applied a Creative Commons Attribution (CC BY) licence to any Author Accepted Manuscript version arising from this submission.

## Author Contributions

**Conceptualization:** Christopher Burton, Helen Dawes, Caroline Dalton.

**Data curation:** Simon Goodwill, Michael Thelwell.

**Formal analysis:** Christopher Burton, Helen Dawes.

**Investigation:** Christopher Burton, Helen Dawes, Caroline Dalton.

**Methodology:** Christopher Burton, Helen Dawes, Caroline Dalton.

**Project administration:** Caroline Dalton.

**Resources:** Simon Goodwill, Michael Thelwell, Caroline Dalton.

**Software:** Simon Goodwill, Michael Thelwell.

**Supervision:** Helen Dawes, Caroline Dalton.

**Validation:** Simon Goodwill, Michael Thelwell.

**Visualization:** Christopher Burton.

**Writing – original draft:** Christopher Burton, Helen Dawes, Caroline Dalton.

**Writing – review & editing:** Christopher Burton, Helen Dawes, Simon Goodwill, Michael Thelwell, Caroline Dalton.

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
