## [Decision Letter · Decision Letter 0]

18 Aug 2022

PONE-D-22-17123

Symptom variation, correlations, and relationship to physical activity in Long Covid: intensive longitudinal study

PLOS ONE

Dear Dr. Burton,

Thank you for submitting your manuscript to PLOS ONE. After careful consideration, we feel that it has merit but does not fully meet PLOS ONE’s publication criteria as it currently stands. Therefore, we invite you to submit a revised version of the manuscript that addresses the points raised during the review process.

We look forward to receiving your revised manuscript.

Kind regards,

Ioannis G. Fatouros

Academic Editor

PLOS ONE

Journal Requirements:

    "No authors have competing interests"

4. Please include a caption for figure 3.

Reviewers' comments:

Reviewer's Responses to Questions

**Comments to the Author**

1. Is the manuscript technically sound, and do the data support the conclusions?

Reviewer #1: Yes

Reviewer #2: Partly

2. Has the statistical analysis been performed appropriately and rigorously? 

Reviewer #1: Yes

Reviewer #2: Yes

3. Have the authors made all data underlying the findings in their manuscript fully available?

Reviewer #1: Yes

Reviewer #2: Yes

4. Is the manuscript presented in an intelligible fashion and written in standard English?

Reviewer #1: Yes

Reviewer #2: Yes

5. Review Comments to the Author

Reviewer #1: The manuscript was enjoyable to read. I would like to address some points that might be confusing.

1. The title should be more representative of the results. For instance, symptom assessment was based on perceived evaluation by the participants subjectively. Additionally, due to the lack of correlations, perhaps the term of interoception, as a potential explanation of the results, should be added in the title.

2. A figure depicting study design would be nice.

3. I see that fatigue was the main symptom. However, I haven’t understood how the rest of the symptoms were measured and expressed independently of fatigue. Fatigue is a broad and vague symptom with multifactorial origins. It may affect the interoception of the rest of the symptoms examined. How was fatigue distinguished from other post-COVID-19 symptoms, independently? In other words, how the possibility of a confounding intercorrelation was addressed/excluded? Is it possibly implied that fatigue amplifies the expression of the rest of the symptoms?

Towards the same direction goes the lack of activity. Exercise is thought to intervene with the information processing and the output of the interoceptive system.

4. 57 citation should be excluded as it has not been peer-reviewed – despite the clear indication provided.

5. Please, proofread as some typographic mistakes were found.

Reviewer #2: This is an excellent study with high clinical significance.

My comments are listed below:

Major concerns: Even though in the current study’s title is stated the word “long covid” the study design has included only 27 participants described having a positive PCR test. It is stated clearly that the analysis of variance showed no significant difference in baseline measures between those testing positive for Covid-19, negative or not tested, however, it is misleading to discuss about Long Covid Symptoms when only the 33% of the participants were verified covid patients. The authors need to address this issue by either changing the title of the manuscript or removing all non-tested participants (or running a separate analysis including only those with positive test).

Minor: I was wondering if the authors have considered the lack of commitment to comply with the researchers’ instructions in collecting the data as one of the symptoms of Long covid? Please elaborate further in the manuscript the fact that the adherence rate was moderate.

Minor: What about the rate of vaccination among the participants? Any relation to the severity of the symptoms? Please response.

6. PLOS authors have the option to publish the peer review history of their article (what does this mean?). If published, this will include your full peer review and any attached files.

Reviewer #1: No

Reviewer #2: **Yes: **Giorgos K. Sakkas

---

## [Author Response · Author response to Decision Letter 0]

11 Nov 2022

Thank you for the constructive and thoughtful comments of your reviewers. We are pleased that the found the manuscript to be of high quality, clinical importance and enjoyable to read.

They both raise important points and we have responded to them in sequence.

Reviewer #1: The manuscript was enjoyable to read. I would like to address some points that might be confusing.

1. The title should be more representative of the results. For instance, symptom assessment was based on perceived evaluation by the participants subjectively. Additionally, due to the lack of correlations, perhaps the term of interoception, as a potential explanation of the results, should be added in the title.

Thanks for encouraging us to think again about this. We have changed it to “Within and between-day variation and associations of symptoms in Long Covid: intensive longitudinal study.” We have chosen this in order to emphasise the time-scales on which this data was collected and analysed (which is what differentiates it from other studies). From this, we think that it follows that the symptoms are a subjective experience. We have not included interoception in the title because (a) we didn’t formally test it, and (b) our patient advisors felt it was imposing an answer prematurely. 

2. A figure depicting study design would be nice.

We have added this 

3. I see that fatigue was the main symptom. However, I haven’t understood how the rest of the symptoms were measured and expressed independently of fatigue. Fatigue is a broad and vague symptom with multifactorial origins. It may affect the interoception of the rest of the symptoms examined. How was fatigue distinguished from other post-COVID-19 symptoms, independently? In other words, how the possibility of a confounding intercorrelation was addressed/excluded? 

The point about fatigue and its intersection with other symptoms is well made, indeed this is one of the issues that this work exposes. However, in most accounts of lived experience in this condition, fatigue is described as a distinct thing and we are cautious about challenging this subjective experience so bluntly. On the other hand, as the reviewer points out, the data show that fatigue is strongly correlated with other symptoms. 

We should have said more in the original manuscript about the development and pilot study. We checked with people with lived experience, during initial app development and after our pilot study to ensure that the app made sense and that they felt able to distinguish the different symptoms. There were no reported problems. We have added a sentence to this effect in the Methods / Study Design section.

We could also have been clearer about addressing this issue by reporting both adjusted and unadjusted (partial) correlations. We have edited the Overview of Analysis subsection to describe this. 

Is it possibly implied that fatigue amplifies the expression of the rest of the symptoms?

That is one possible explanation. Equally if fatigue is the “feeling that occurs when voluntary action should stop or not be started” then it is highly likely that other symptoms will trigger a state of fatigue. At this stage we know so little about fatigue that we prefer to keep an open mind. 

Towards the same direction goes the lack of activity. Exercise is thought to intervene with the information processing and the output of the interoceptive system.

We are not saying that it doesn’t. But in this research, we were focused on short-term associations not on longer term changes. If we take an altered interoception approach to interpreting this condition, then short term increases in activity will create more noise in the interoceptive system without changing the predictive priors the brain uses. Any change in priors probably takes time or perhaps specific disruption (which is perhaps where anecdotal things like cold water swimming might come in).

4. 57 citation should be excluded as it has not been peer-reviewed – despite the clear indication provided.

We have found increasing citation in journals of material on public preprint servers. Before deleting this, we would prefer to check with the editorial team.

5. Please, proofread as some typographic mistakes were found.

We have done this and apologise for missing these in the original submission.

Reviewer #2: This is an excellent study with high clinical significance.

My comments are listed below:

Major concerns: Even though in the current study’s title is stated the word “long covid” the study design has included only 27 participants described having a positive PCR test. It is stated clearly that the analysis of variance showed no significant difference in baseline measures between those testing positive for Covid-19, negative or not tested, however, it is misleading to discuss about Long Covid Symptoms when only the 33% of the participants were verified covid patients. The authors need to address this issue by either changing the title of the manuscript or removing all non-tested participants (or running a separate analysis including only those with positive test).

Thanks for this comment which reflects on important issues both of defining the illness and of the issue of whose illness this actually is.

We agree that more information could have been included and we have added additional data about the comparison between groups in our baseline measures, visual analogue scales and accelerometery. We have made additions to methods and results and added an additional figure. However, we found little difference between the PCR groups in any of the data we collected / analysed (the exception, as always, being loss of taste / smell).

However, we disagree with the reviewer that it is “misleading” to include people without a positive PCR and in this we are strongly supported by our patient collaborators. Firstly one third of cases had their acute illness in the early stages of the pandemic when PCR testing was not available for community cases. Second, we know from a large French study that while approximately half of people with the post covid condition did not have a positive PCR, at least half of these had positive antibodies (before vaccination). On balance, we believe that it is not appropriate in the current state of knowledge to exclude people on the basis of a limited availability / once-off PCR test and certainly our patient collaborators hold very strongly to this view. 

Minor: I was wondering if the authors have considered the lack of commitment to comply with the researchers’ instructions in collecting the data as one of the symptoms of Long covid? Please elaborate further in the manuscript the fact that the adherence rate was moderate.

We had considered something like this. In the results we note that “There was no association between number of completed entries and mean self-reported fatigue (spearman’s rho -0.05)”. 

We have now added a sentence to the discussion indicating that completion rates (78% of possible entries) were comparable to those expected in a definitive text on the intensive longitudinal method (80%).

Minor: What about the rate of vaccination among the participants? Any relation to the severity of the symptoms? Please response.

We have added a sentence to the Participant Characteristics section that because the large majority had become unwell before immunisation was available, we did not collect that data.

---

## [Editor Report · Decision Letter 1]

27 Dec 2022

Within and between-day variation and associations of symptoms in Long Covid: intensive longitudinal study.

PONE-D-22-17123R1

Dear Dr. Burton,

We’re pleased to inform you that your manuscript has been judged scientifically suitable for publication and will be formally accepted for publication once it meets all outstanding technical requirements.

Kind regards,

Ioannis G. Fatouros

Academic Editor

PLOS ONE

Additional Editor Comments (optional):

Dear Authors,

Unfortunately both reviewers have not responded to my second call. As such, I moved ahead and I reviewed your manuscript my self. It appears that you have addressed all of the reviewers' comments successfuly and you have improved your manuscript considerably. However, I agree with the comment raised by Reviewer 2 on deleting citation 57. As such, I find your manuscript suitable for publication but you need to eliminate reference 57.

Sincerely,

Ioannis G. Fatouros
---

## [Editor Report · Acceptance letter]

6 Jan 2023

PONE-D-22-17123R1 

Within and between-day variation and associations of symptoms in Long Covid: intensive longitudinal study. 

Dear Dr. Burton:

I'm pleased to inform you that your manuscript has been deemed suitable for publication in PLOS ONE. Congratulations! Your manuscript is now with our production department. 

Kind regards, 

on behalf of

Dr. Ioannis G. Fatouros 

Academic Editor

PLOS ONE